# Teaching Approaches and Educational Technologies in Teaching Mathematics in Higher Education

**Renate Nantschev** [1,*], **Eva Feuerstein** [2], **Rodrigo Trujillo González** [3], **Israel Garcia Alonso** [3], **Werner O. Hackl** [1], **Konstantinos Petridis** [4,5], **Evangelia Triantafyllou** [6] and **Elske Ammenwerth** [1]

1   Institute of Medical Informatics, UMIT – Private University for Health Sciences, Medical Informatics and Technology, Eduard Wallnöfer-Zentrum 1, 6060 Hall in Tirol, Austria; werner.hackl@umit.at (W.O.H.); elske.ammenwerth@umit.at (E.A.)
2   Faculty of Biomedical Engineering, Czech Technical University, Nám. Sítná 3105, 272 01 Kladno, Czech Republic; neumaeva@fbmi.cvut.cz
3   Department of Mathematical Analysis, University of La Laguna, 38071 San Cristóbal de La Laguna, Spain; rotrujil@ull.edu.es (R.T.G.); igarcial@ull.edu.es (I.G.A.)
4   Department of Electronic Engineering, Hellenic Mediterranean University, 73100 Chania, Crete, Greece; cpetridis@hmu.gr
5   The European University ATHENA, 71410 Heraklion, Crete, Greece
6   Department of Architecture, Design, and Media Technology, Aalborg University, A. C. Meyers Vænge 15, 2450 Copenhagen SV, Denmark; evt@create.aau.dk
*   Correspondence: renate.nantschev@umit.at

**Abstract:** The growing use of technology for mathematics in higher education opens new pedagogical and technological challenges for teachers. The objective of this study was to analyze the teaching approaches and technology-related pedagogical competencies of 29 mathematics teachers (15 females and 14 males) from nine European countries. After conducting semi-structured interviews, the Approaches to Teaching Inventory (ATI-16) and the Technological Pedagogical Content Knowledge (TPACK) framework survey were applied. The results show large individual variations in teaching approaches, technological competencies, and institutional support. One-third of teachers apply a more student-centered approach, one-third a more teacher-centered approach, and one-third a mixed approach. Educating and supporting teachers in embracing educational technologies thus needs to be tailored strongly to individual needs and the available institutional support resources and infrastructure.

**Keywords:** approaches to teaching; educational technologies; mathematics education; quantitative study

## 1. Introduction

For centuries, mathematics has been associated with traditional blackboard teaching. The current situation with the COVID-19 pandemic emphasizes the need for mathematics teachers of higher education to move to online teaching. Educational institutions are thus advised to provide adequate training for teachers to improve their online teaching skills [1].

Nowadays, teaching mathematics requires not only content knowledge (i.e., knowing what to teach) but also pedagogical knowledge (i.e., knowing how to teach and how to deal with students' problems and learning difficulties) [2]. Additionally, teachers should have knowledge of the appropriate educational technologies and their potential.

The use of technologies in mathematics teaching and learning can be classified in two dimensions: The use of domain-specific mathematics software (e.g., GeoGebra, https://www.geogebra.org/); and the

general use of learning technologies (e.g., Moodle, https://moodle.org/) [3]. Mathematics-specific software applications are tools with the ability to increase students' conceptual understanding mathematical modeling, visualization and simulation [4].

Education and technology have become two interdependent concepts in mathematics education [5]. Consequently, a low level of digital literacy among mathematics teachers may hinder the adoption of modern pedagogical and technological approaches in mathematics learning and teaching [3]. Many teachers are beginning to adopt the perspective that learning can only be successful if learners construct their knowledge and do not merely extend their knowledge through memorization [5]. Teachers should therefore help students to use and reinforce the knowledge they have and to produce new knowledge. When they do, the teaching approach shifts from a teacher-centered to a more student-centered approach. Studies indicate that the integration of technology in teaching has the potential to move teaching towards a more student-centered approach [6]. Calls for reforming mathematics education by considering more innovative teaching approaches are often rooted in constructivist theory [3]. Student-centered teaching approaches play an essential role in this process. Several such teaching approaches (e.g., Problem-Based Learning) have been developed and have started to gain momentum in mathematics education [3].

To foster innovative pedagogical and technological approaches in mathematics education, the Erasmus + Capacity Building in the Field of Higher Education (CBHE) project "Innovative Teaching Education in Mathematics" (ITEM) was launched in 2018 [7]. It aims at improving mathematics teaching practice in higher education by applying innovative instructional approaches. Mathematics is essential in addressing major challenges in science, technology, and engineering. The main objectives of ITEM are to increase students' motivation to build their mathematical skills and to raise students' success rate in mathematics courses in engineering education. To achieve these objectives, the project aims to improve teaching quality, enhance the support provided to students to overcome their difficulties and misconceptions of how to learn mathematics, and enrich the teachers' toolbox with engaging teaching techniques. The ITEM project team consists of 16 partners (including ten test institutions) from higher education in ten different countries (Austria, Czech Republic, Denmark, Greece, Israel, Kosovo, North Macedonia, Spain, Sweden, Uzbekistan).

This paper presents results from studies conducted within the first ITEM phase, focusing on diagnosis of the teaching approaches and technology-related pedagogical competencies of the math teachers involved, as a precondition for the design of the training programs for mathematics teachers. Here, the ITEM project has started to plan and implement workshops for mathematics teachers at the ITEM consortium institutions to train them in student-centered methods such as Problem-Based Learning (PBL) and Project-Oriented Problem-Based Learning (POPBL).

Problem-Based Learning (PBL) is an active pedagogical student-centered learning method. It is a shift away from the traditional educational practice in which the teacher is at the center. PBL is an "instructional (and curricular) student-centered approach that empowers students to conduct research, integrate theory and practice, and apply knowledge and skills to develop a viable solution to a defined problem" [8]. The central element of PBL is a real-world problem that helps to initiate the learning process. The assumption is that students are triggered to learn when faced with a problem that cannot be resolved with the knowledge they have. Since the students do not know the solution to the problem, they start looking for new sources of information and knowledge to solve it. Another aspect is that students learn from each other; therefore, working in groups is another fundamental aspect of PBL [8].

In addition to targeted teaching methods, ITEM aims to train teachers in the use of educational technologies (learning management systems, dynamic learning tools such as GeoGebra, Desmos, etc.) related to mathematics teaching.

To describe what teachers need to know to successfully integrate technology into teaching mathematics, the Technological Pedagogical Content Knowledge (TPACK) framework can be used.

The TPACK framework comprises seven domains as shown in Figure 1: The Technological Knowledge (TK), Pedagogical Knowledge (PK), Content Knowledge (CK), Technological Content

Knowledge (TCK), Technological Pedagogical Knowledge (TPK), and Technological Pedagogical Content Knowledge (TPACK) [9].

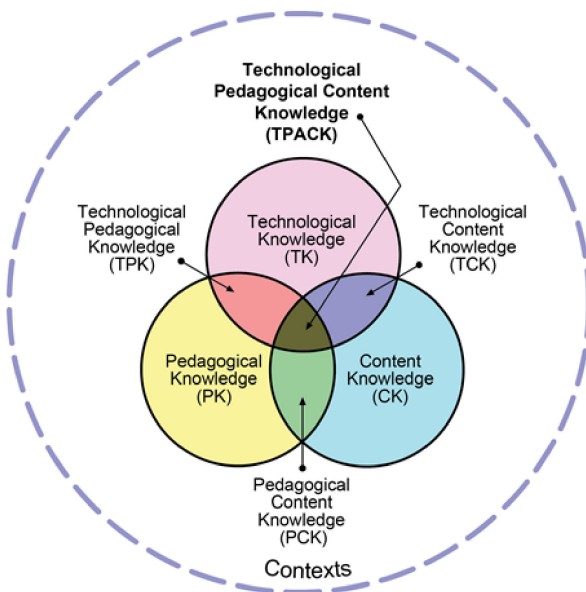

**Figure 1.** The Technological Pedagogical Content Knowledge (TPACK) Framework. Reproduced by permission of the publisher, © 2020 by tpack.org. [10].

- **Content Knowledge (CK):** Content knowledge is about learning and teaching the subject matter in the field to be taught;
- **Pedagogical Knowledge (PK):** Teacher knowledge of instructional methods, strategies, and techniques used in the classroom;
- **Technological Knowledge (TK):** Teacher knowledge of educational technologies and technical infrastructures (software, hardware, internet). Technology knowledge refers to the information and skills that teachers should have with regard to using technology;
- **Pedagogical Content Knowledge (PCK):** Teacher knowledge of the effective use of teaching methods and techniques to teach the course contents. The teacher knows how students learn and how to measure and evaluate their learning outcome;
- **Technological Content Knowledge (TCK):** Teacher knowledge of the use of appropriate educational technology for teaching the course content;
- **Technological Pedagogical Knowledge (TPK):** Teacher knowledge of effective pedagogical use of educational technology. Technological pedagogical knowledge refers to knowledge of tools and their functionalities and of the interrelation between tools and tasks;
- **Technological Pedagogical Content Knowledge (TPACK):** Teacher knowledge of the effective combination of pedagogy and educational technology for teaching mathematical content.

TPACK has been successfully used as a framework to analyze and improve science and mathematics teaching in higher education [11,12].

In this paper, we aim to explore teaching approaches and technology-related pedagogical competencies among mathematics teachers involved in the ITEM project to be able to prepare well-tailored training programs for mathematics teachers within ITEM. The research questions of this study were:

Q1: Which teaching approaches are applied by mathematics teachers in higher education?

Q2: From the perspective of TPACK, which content-related and technological knowledge do the mathematics teachers show?

Q3: Based on the results from Q1 and Q2, which areas should be covered in the workshops for ITEM teachers to better prepare the mathematics teachers for student-centered teaching?

## 2. Methods and Materials

This study took a quantitative approach to answer the study questions. We used two standardized surveys: the Approaches to Teaching Inventory (ATI) survey and a TPACK survey. Both questionnaire are available as Supplementary Materials. The quantitative survey aimed at attaining comparable data among participating teachers and serves as a pre-measurement for later planned post-measurements after completion of the teachers' training programs within the ITEM project. To obtain context information on the recent use of technology in mathematics teaching and on teaching approaches, we first conducted semi-structured interviews with senior mathematic teachers of all participating ITEM institutions.

### 2.1. Interviews on Technology Use and Teaching Approaches in Mathematics Education

**Participants:** We conducted semi-structured interviews with senior mathematic teachers in the participating ITEM institutions of higher education. Overall, ten mathematics teachers, four women and six men, from ten higher education institutions in seven countries participated in the interviews. The seven countries were: Kosovo, Uzbekistan, Greece, North Macedonia, Israel, Czech Republic, Sweden. All teachers were responsible for teaching linear algebra and calculus in the first and second semester in different engineering study programs. The interviews were conducted via online video calls between June and September 2019. The duration of interviews was between 15 and 35 min.

**Instruments and procedure:** We developed a semi-structured interview guideline. The interview guideline focused on two main topics. The first topic was the use of technology in teaching mathematics and the available technical infrastructure. For example, some of the questions focusing on this topic were: Do you use special software for teaching mathematics? And if so, which and in which form? Are the students invited to use their smartphones, tablets, or laptops in your classroom? If yes, for which purpose? Does your classroom offer wifi? Second, the interviews focused on the use of student-centered teaching approaches. Some of the questions focusing on this topic were: Do you use real-life examples and problems from other fields (e.g., biology, economics, physics) to teach mathematics? Are there elective activities where your students have to do research on selected mathematics topics?

**Data analysis**: Interviews were transcribed verbatim and summarized based on the structure of the interview guideline.

### 2.2. Quantitative Standardized ATI and TPACK Survey

**Participants:** All mathematics teachers within the ITEM project who were nominated to participate in the proposed ITEM training workshops were invited to the survey. Overall, 61 teachers from 13 different universities in nine countries were invited to fill in the online survey between August and September 2019. Participants comprised senior lecturers, assistant professors, associate professors, and full professors. The survey was conducted before the first ITEM teacher training workshop took place. Participants were asked to answer both surveys by considering the most typical mathematics course they teach. Overall, 29 mathematics teachers from nine countries responded to the survey (response rate: 48%), with a female/male ratio of 15:14. With respect to years of teaching experience, the distribution is as follows: 27% of respondents have more than 20 years teaching experience, 24% between 15 and 19 years, 21% between 10 and 14 years, 14% between five and nine years, and 14% less than five years, spanning between one and four years.

**Instruments and procedure:** We used the Approaches to Teaching Inventory (ATI-16), a validated instrument with 16 items that allows us to measure two opposing approaches to teaching: the Information Transfer/Teacher-Focused (ITTF) approach and the Conceptual Change/Student-Focused (CCSF) approach [13,14]. Scale reliabilities (Cronbach's alphas) for ITTF

and CCSF scale (ATI-16) is 0.539 and 0.505, respectively [15]. The ITTF approach corresponds to a more teacher-centered strategy, while the CCSF approach corresponds to a more student-centered strategy [13].

To assess the TPACK dimensions of technological, pedagogical, and content knowledge, we used a modified version of the HE-TPACK instrument [16]. This instrument measures the TPACK level in higher education [17]. Originally, this instrument contained 57 items. The modified version we used consisted of 26 items, including demographic information.

Both surveys were answered on a five-point Likert scale ranging from 1 (only rarely true) to 5 (almost always true).

**Data analysis:** For both the ATI CSSF subscale and the ATI ITTF subscale, we calculated the mean of the respective items for each respondent as well as the difference between the two means. For the TPACK survey, we used descriptive statistics for all items.

## 3. Results

We first present the results of the semi-structured interviews on using technology in mathematic teaching. The interview results are summarized in two main categories: use of technologies in teaching mathematics, and teaching approaches in mathematics. Second, we present the results of the Approaches to teaching inventory (ATI) and Technological, Pedagogical, Content Knowledge (TPACK) survey.

### 3.1. Recent Use of Technology in Teaching Mathematics

In the first part of the interviews, we focused on the technologies the teachers use in their classes. Regarding general learning technologies, two (of the seven participating institutions for interviews) use Moodle as a learning management system (LMS). The other institutions use different LMSs (e.g., eClass, Canvas) and one institution does not use any kind of LMS. Most of the teachers indicated they use the platforms only to provide slides for the upcoming lessons. (*"I scan my handout and give them a file. They can submit their hand-in tasks."*; *"For the materials, we use Google Drive."*) Only one teacher uses the learning platform as an interactive tool with students. A university-wide strategy in using an online learning platform is not mentioned among any of the interviewed teachers.

Regarding domain-specific mathematics software, nearly all interviewed teachers use special mathematics software to support learning. Often-mentioned tools include Matlab, GeoGebra, Desmos and Derivative Calculator. (*"To use tools in the lectures to illustrate things. We use that to draw things and drafts. To draw pictures on the board. We use it also for hand-in assignments."*; *"It is a quite useful thing. The students use it also-we try to encourage them."*)

All teachers have WiFi available in their classroom. Students mostly bring their private devices and use them in class. Only one university does not allow the use of smartphones or tablets in class.

### 3.2. Teaching Approaches in Mathematics

This second part of the interviews focused on using student-centered approaches in teaching, especially on using Problem-Based Learning (PBL) based on real-life examples. Nearly all teachers are already attempting to work with real-life examples, especially from physics. Only a few use examples from electronics or engineering. One teacher has the opportunity to work with examples from industry partners: *"In the practice part, the students have to solve real-life questions in teams of four to five."*

Most teachers state that they use real-life examples to capture the interest of their students and to show how mathematics is applied to real-world problems: *"So it is a kind of entertaining math. Because students think that math is difficult."*; *"We are all changing the way we teach mathematics. We need applied mathematics for our students."* In the majority of cases, the teacher prepares a list of real-life examples and the students can then choose which one they want to solve.

We also asked the teachers whether they consider their teaching approach to be more teacher-focused or more student-focused. Most of the teachers described their approach as

student-focused. They explained that they try to encourage students to discuss topics during the lectures, or that they motivate students to work together in smaller groups to improve learning. Fewer teachers considered their approach to be more teacher-focused, with a focus on the presentation of content: *"The goal is to deliver math courses."*

### 3.3. Approaches to Teaching Inventory (ATI)

Figure 2 shows the aggregated results of the student-centered subscale (CCSF) and the teacher-centered subscale (ITTF) of the ATI survey for all 29 teachers. The range for the Conceptual Change/Student-Focused (CCSF) subscale is 1.0 to 4.4. The range for the Information Transfer/Teacher-Focused (ITTF) subscale is 2.0 to 4.3.

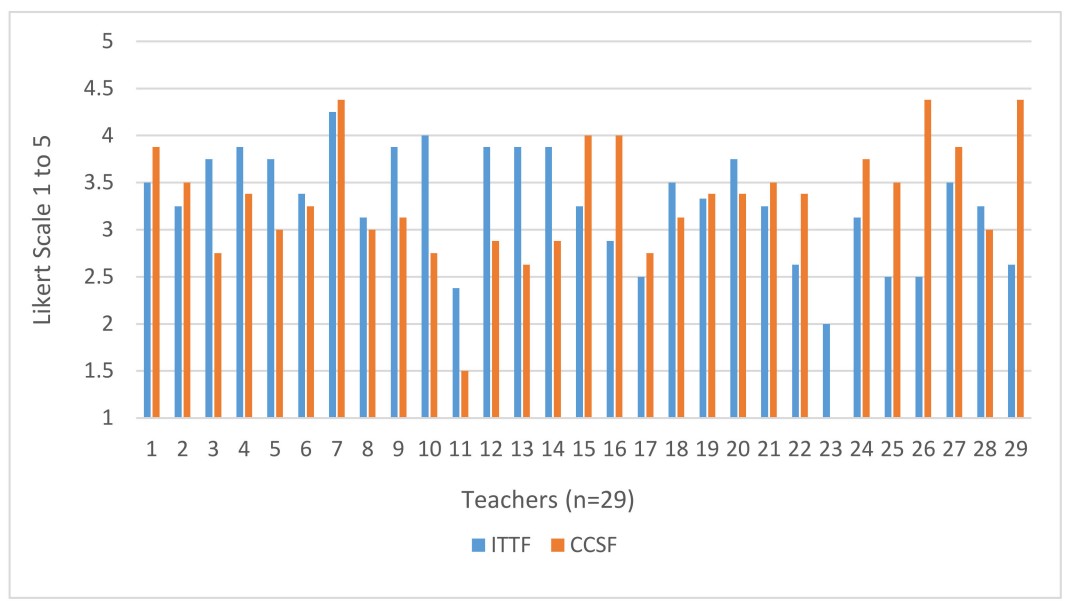

**Figure 2.** Score of each teacher (*n* = 29) in the category Information Transfer/Teacher-Focused approach (ITTF) and Conceptual Change/Student-Focused approach (CCSF) of the Approaches to Teaching Inventory (ATI); 1 = minimum, 5 = maximum.

To better understand the overall teaching approach of each teacher, we subtracted the ITTF score from the CCSF score (Figure 3). For example, teacher 10 has an ITTF score of 4 and a CCSF score of 2.75. The overall score is $2.75 - 4 = -1.25$, which is displayed in Figure 2. The higher the calculated score, the higher the student-centeredness of an individual teacher. Negative values indicate a more teacher-focused approach, positive values a more student-focused approach.

The quantitative clustering of the results in Figure 3 shows three general clusters: first, teachers with a stronger teacher-focused approach (left part of Figure 3); second, teachers with a balanced approach (middle part of Figure 3); and third, teachers with a stronger student-focused approach (right part of Figure 3).

These results show large individual variations in teaching approaches. To see whether the institution plays a role in the teaching approach, we compared results of the institutions participating in the survey. No meaningful patterns could be found in this sub-analysis.

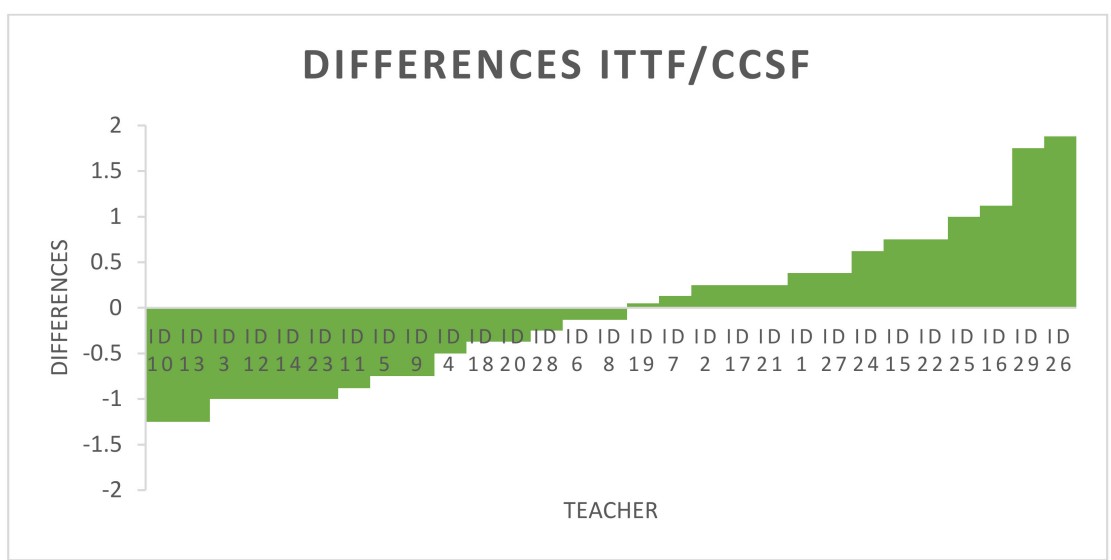

**Figure 3.** Differences of each teacher (*n* = 29) between the ITTF approach and CCSF approach. Negative values indicate a more teacher-centered approach. Positive values indicate a more student-centered approach.

*3.4. Technological, Pedagogical, Content Knowledge (TPACK)*

Figures 4 and 5 present the results of the TPACK survey for all 29 respondents.

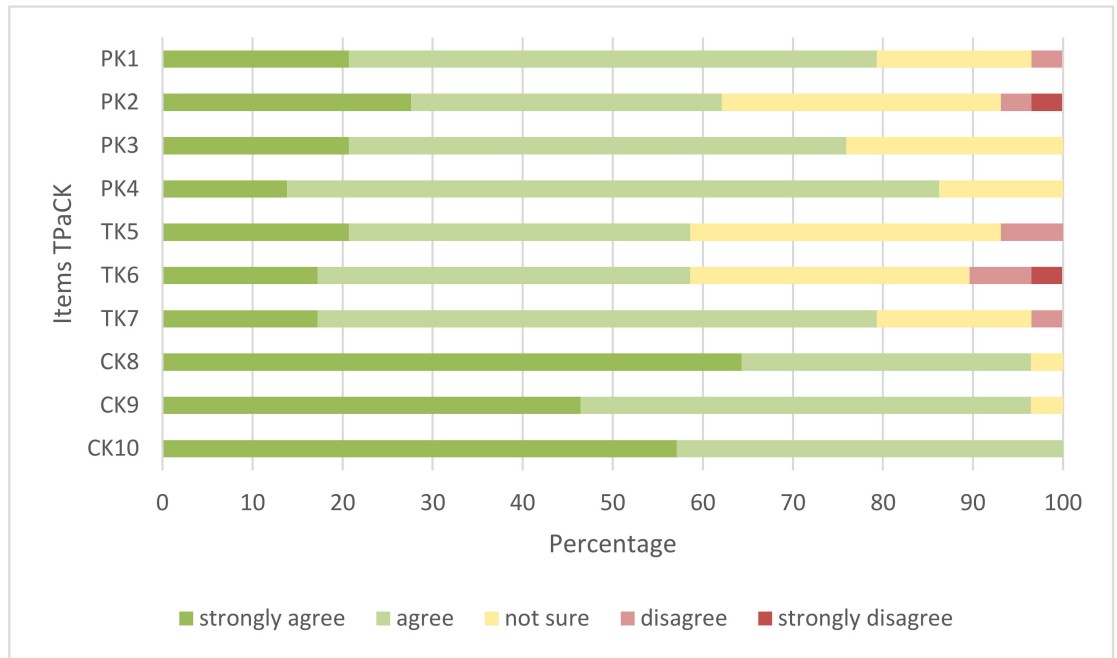

**Figure 4.** Results in percent for each item of TPACK questions 1 to 10 (*n* = 29).

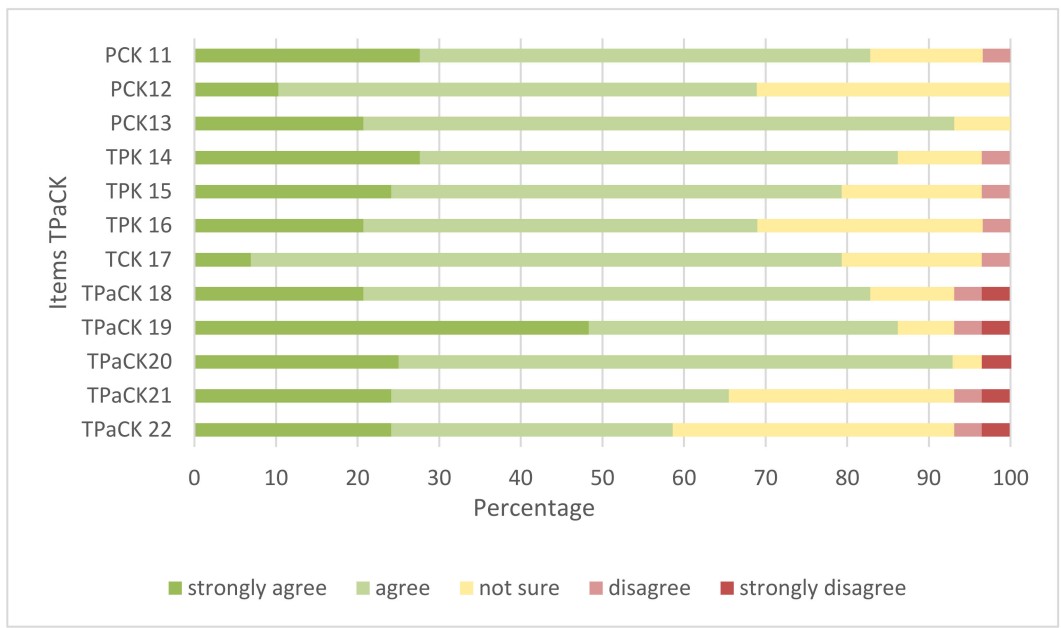

**Figure 5.** Results in percent for each item of TPACK questions 11 to 22 (*n* = 29).

Survey items for Figure 4:

PK 1: I have a clear understanding of pedagogy (e.g., designing instruction, assessing students' learning);

PK 2: I am familiar with a wide range of practices, strategies, and methods that I use in my teaching;

PK 3: I know how to assess student learning;

PK 4: I know how to motivate students to learn;

TK 5: I am familiar with a variety of hardware, software, and technology tools that I can use for teaching;

TK 6: I know how to troubleshoot technology problems when they arise;

TK 7: I can decide when technology can be beneficial to achieving a learning objective;

CK 8: I have a comprehensive understanding of the curriculum I teach;

CK 9: I explain to students the value of knowing concepts in my discipline;

CK 10: I make connections between the different topics in my discipline.

Survey items for Figure 5:

PCK 11: I understand that there is a relationship between content and the teaching methods used to teach that content;

PCK 12: I can anticipate and address students' preconceptions and misconceptions;

PCK 13: I understand what topics or concepts are easy or difficult to learn;

TPK 14: I understand how teaching and learning change when certain technologies are used;

TPK 15: I understand how technology can be integrated into teaching and learning to help students achieve specific pedagogical goals;

TPK 16: I know how to be flexible with my use of technology to support teaching and learning;

TCK 17: I understand how the choice of technologies allows and limits the types of content ideas that can be taught;

TPACK 18: I integrate educational technologies to increase student opportunities for interaction with ideas

TPACK 19: I motivate my students to use learning technologies to support their individual learning;

TPACK 20: I understand what makes certain concepts difficult to learn for students and how technology can be used to leverage that knowledge to improve student learning;

TPACK 21: I understand how to integrate technology to build upon students' prior knowledge of curriculum content;

TPACK 22: I know how to operate classroom technologies and can incorporate them into my particular discipline to enhance student learning.

We clustered the seven questions with the highest rate of "strongly agree" and "agree" responses and the seven questions with the highest rate of "disagree" and "strongly disagree" responses. Details are shown in Tables 1 and 2.

**Table 1.** Items of the TPACK survey that were most often rated as agree and strongly agree (n = 29).

| Ranking | ITEM | Agree/ Strongly Agree | TPACK Questions |
|---|---|---|---|
| 1 | CK10 | 100% | I make connections between the different topics in my discipline. |
| 2 | CK9 | 96% | I explain to students the value of knowing concepts in my discipline. |
| 3 | CK8 | 96% | I have a comprehensive understanding of the curriculum I teach. |
| 4 | PCK13 | 93% | I understand what topics or concepts are easy or difficult to learn. |
| 5 | TPACK20 | 93% | I understand what makes certain concepts difficult to learn for students and how technology can be used to leverage that knowledge to improve student learning. |
| 6 | PK4 | 86% | I know how to motivate students to learn. |
| 7 | TPK 14 | 86% | I understand how teaching and learning change when certain technologies are used. |

**Table 2.** Items of the TPACK survey that were most often rated disagree and strongly disagree (n = 29).

| Ranking | ITEM | Disagree/ Strongly Disagree | TPACK Question |
|---|---|---|---|
| 1 | TK6 | 10% | I know how to troubleshoot technology problems when they arise. |
| 2 | PK2 | 7% | I am familiar with a wide range of practices, strategies, and methods that I use in my teaching. |
| 3 | TK5 | 7% | I am familiar with a variety of hardware, software and technology tools that I can use for teaching. |
| 4 | TPACK 18 | 7% | I integrate educational technologies to increase student opportunities for interaction with ideas. |
| 5 | TPACK 19 | 7% | I motivate my students to use learning technologies to support their individual learning. |
| 6 | TPACK21 | 7% | I understand how to integrate technology to build upon students' prior knowledge of curriculum content. |
| 7 | TPACK 22 | 7% | I know how to operate classroom technologies and can incorporate them into my particular discipline to enhance student learning. |

## 4. Discussion

We analyzed the teaching approaches and the technology pedagogical content knowledge of 29 mathematics teachers from ten higher education institutions in nine countries. In the interviews that were conducted before the survey, the majority of teachers indicated they use a more student-focused approach. The ATI survey showed, however, large individual variations, with around one-third of teachers applying a more student-centered approach, one-third applying a more teacher-centered approach, and one-third applying a mixed approach. These results confirm findings from other studies that also found varying approaches to learning and various levels of TPACK in higher education

teachers, irrespective of type of university [12,18]. As an indicator of a student-centered approach, most interviewed teachers said they use real-life problems, mostly from physics.

During the interviews, when discussing teaching approaches, we had the impression that many teachers had not really thought about their approach. For example, when asked whether their teaching is more student-centered or more teacher-centered, teachers answered that they have a student-based evaluation of their courses and thus consider their approach to be student-centered. The standardized ATI survey provides a more precise and standardized picture of the student-centeredness versus teacher-centeredness of the respondents and shows large variety in student-centeredness.

The TPACK survey revealed high approval for content-related items but lower approval for technology-related items. The interviews showed that learning platforms are mostly used for sharing materials, but almost never for interactive or self-regulated activities. While WiFi is available in all classes, it is mostly used by students for following the lecture but not used by the teacher for active learning in the classroom. Overall, in our study, the support of TPACK items that address the interaction of technology, pedagogy, and content is lower than items on pedagogical and content knowledge alone. While a direct comparison of TPACK levels from different studies is difficult, as studies use varying or modified TPACK surveys, earlier research also shows the same trend, with higher pedagogical and content-related knowledge areas and lower technology-related TPACK knowledge areas [12].

Based on these results, we see a strong need for continuous training of mathematics teachers in two fields: first, the use of technology in teaching, especially for fostering student-centered teaching, and second, the promotion of mathematics teachers' awareness of current technologies that can support content-related pedagogy. These findings are already discussed within the ITEM consortium and will help to shape future teacher training workshops. A recent review of meta-analyses has indeed shown that TPK, TCK, and TPACK show the highest effect sizes for effectively using technology in the mathematics classroom and should thus be considered in teachers' trainings [4]. Other studies have shown that teachers' training can indeed improve TPACK levels [12].

As a limitation, the study input comes only from teachers already involved in the ITEM project. As their teaching approaches and institutional contexts are quite diverse, we believe that our findings might also be relevant for other higher education institutions. However, given the convenience sampling that we applied, these results cannot be seen as representative of the institutions involved or for mathematics teaching in general.

In this study, we had expected to see some patterns of teaching approaches between institutions, but the sample size was too small to derive meaningful patterns.

For this study, we developed a new approach to subtract the ATI student-centered scale from the ATI teacher-centered scale to get a better picture of the individual teaching approach. We were not able to validate this approach based on the ATI literature. Such a formal validation of the resulting score should therefore be performed in a future study.

Training workshops for ITEM teachers are now being planned within the ITEM project. We plan to repeat the ATI and TPACK survey with the same teachers upon completion of the training. We would expect an increase in student-centered approaches and in technological as well as technological pedagogical knowledge. Indeed, earlier research has found that tailored training for teachers can significantly enhance the integrated TPACK scores [12] and that pedagogical training can foster a more student-centered approach [19].

## 5. Conclusions

Our results from a survey of mathematics teacher from seven higher education institutions show large individual variations in teaching approaches, technological competencies, and institutional support. Educating and supporting teachers in embracing educational technologies thus needs to be tailored strongly to the individual needs and the available institutional support resources and infrastructure. The results especially indicate that teacher training needs to focus on competencies in the interaction of technology, pedagogy, and content in mathematics teaching.

**Supplementary Materials:** The following are available online at http://www.mdpi.com/2227-7102/10/12/354/s1: Online Survey ATI 16 and TPACK; Semi-structured Interview Guide.

**Author Contributions:** Conceptualization, R.N.; E.A.; E.F.; R.T.G. and E.T.; methodology, R.N.; E.G.A.; E.F. and E.T.; formal analysis, R.N. and E.A.; investigation, R.N. and W.O.H.; data curation, R.N. and W.O.H.; writing—original draft preparation, R.N. and E.A.; writing—review and editing, E.A.; E.F.; E.T.; R.T.G.; I.G.A.; K.P. and W.O.H.; supervision E.G.A.; project administration R.N. All authors have read and agreed to the published version of the manuscript.

**Funding:** This research has been funded with support from the European Commission (grant number 2018-3265/001-001). This publication reflects the views only of the author, and the Commission cannot be held responsible for any use which may be made of the information contained therein.

**Conflicts of Interest:** The authors declare no conflict of interest.

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
