# Peer review of "Teaching Approaches and Educational Technologies in Teaching Mathematics in Higher Education"

_education, doi:10.3390/educsci10120354_

Round 1

Reviewer 1 Report

This is a manuscript very interesting. I am going to give some feedback in case of being useful for improving some points of this:

TITLE

No comments or considerations. The title is correct.

ABSTRACT AND KEYWORDS

Line 7. Although it is not mandatory, it is recommended to use impersonal verb tenses instead of "we".

It would be interesting to put the number and percentage of women or men in the sample. That way other scientists can get an idea of how representative the sample is in terms of the sex/gender of the participants.

The keywords could be in alphabetical order. At least one of the keywords could be the research design.

The structure is correct.

INTRODUCTION

Good organization of ideas. The introduction is very interesting. However, the theoretical framework needs to be expanded. There are just ten quotes of which 4 are from the last five years. So, the introduction also needs to be updated.

1.2. OBJETIVE

Questions are a good way of guiding research and helping to reflect on what you are trying to achieve. In any case, it would be useful to introduce hypotheses to be later contrasted in the discussion.

MATERIALS AND METHOD

As an empirical investigation through a survey, it should also be shown in subsections:

    • Participants.
    • Instruments.
    • Procedure
    • Analysis of data.

The size of the sample is very small and this implies that it is difficult to generalize the results. It should be recommended to expanse the sample to gain external validity or justify the size in greater depth.

Further psychometric support of the validity of the instrument is needed.

RESULTS

Line 250. Avoid underlying. ídem in the rest of the cases.

Numerous tables and graphs are provided to help understand and illustrate the research. This is a positive aspect.

DISCUSSION

This section needs to be improved.

It would be interesting to explain whether each hypothesis contradicts or is consistent with previous studies.

No bibliographical references have been placed in this section and the discussion should compare the results of the work with others in the theoretical framework. It is necessary to introduce references.

The manuscript raises the limitations of the study which is considered positive.

CONCLUSIONS

This section does not appear. It should appear in this type of study.

REFERENCES

Check the format.

The number of references must be expanded.

To sum up, the theme is interesting. Thank you very much for your work and attention. 

Author Response

Dear Reviewer,

Please see the attachment to our comments. 

Thank you for reviewing our manuscript. 

Reviewer 2 Report

The manuscript is well structured and well written with regard to content. However, there are some formal and language issues.

In the introduction, GeoGebra is mentioned without any reference.

The authors mix style of references. In some cases they use name and year, e.g. Savery (2006) or Trigwell & Prosser (2004), instead of number of the reference as listed in References part.

Several times ATI is mentioned. However, no reference is related to it.

Page 7, lines 245 and 246 - probably a mistake. In line 246 figure number is missing.

Page 8, line 289 - error message - reference source not found.

Section 4 Discussion: another font size is used than in previous sections.

References: item 5 - URL is missing.

English needs revision.

Author Response

(The authors gave the same response as above.)

Reviewer 3 Report

The text of this article is very interesting for the community of educators of mathematics with Educational Technology. The TPACK model is presented as the reference framework of the study as well as in the instruments. The study cannot be generalized due to the small sample but it is useful as a background for possible larger studies. The method should be improved as the mixed methods approach is not suitable in this context or with the results it offers. I could describe it as a quanti-quali methodology due to its descriptive character. The conclusions are in line with the objectives of the work.

Author Response

(The authors gave the same response as above.)

Reviewer 4 Report

Dear author

Regarding the article submitted for review, it is my opinion that it is an article that corresponds to a very initial work.

Title: It is suggested to modify the reference in the title to the "international survey", for a more significant clarification, deleting the reference to the method.

Abstract: The abstract contains the necessary structural elements. In this section and the following, the most appropriate translation of university professor is not teachers but professor.

Introduction: In this section it is necessary to deepen and broaden the state of the question (only 8 references in a background section, and only 12 in the total of the article, and only three from the last 5 years). The descriptive excess of the Erasmus Action in itself does not leave anything to the state of the question about the research problem.Although the objectives are debatable, they are not usually part of the Introduction.

Methods The sample for the quantitative study is not very robust, and especially the analysis procedure chosen to alleviate this weakness is not explained.

Results:Check numbering (for example, it is understood that section 4.4 should be 3.4? Only a fully descriptive account is presented. In the qualitative part there are no textual references or clear presentation of categories, and in the quantitative part there are only tables of percentages that in a certain way make the weakness of the sample little transparent.

Discussion: This section in my opinion presents a serious error of approach. It could be considered a Conclusion, which otherwise does not have it, but is not admissible as a discussion. There is not in the whole section a single reference of contrasts or discussion.

There is no item Conclusion

Regards

Author Response

(The authors gave the same response as above.)

Round 2

Reviewer 1 Report

Thank you for your effort and time in improving the manuscript. 

Author Response

Dear Reviewer,

thank you for your reply and all your suggestions to improve our manuscript. 

Best regards 

Reviewer 4 Report

The authors make a notable effort to improve this new version.
They have proposed another title, improved the introduction and discussion.

However, the article still has relevant deficiencies that I believe can and should be improved.

They self-qualify their research as qualitative and quantitative.
Regarding the qualitative, the characterization of the participants is very weak for a qualitative research. The qualitative results have no relevance, or categorization.

In the quantitative part, it seems that the statistical analysis could be expanded, and not limited only to descriptive statistics.

Therefore, it is recommended to improve the methodology section in relation to the qualitative part, and the qualitative and quantitative results, in the direction indicated.

Author Response

Dear Reviewer,

thank you for your fast feedback and the very helpful comments. We have carefully discussed the comments and tried to revise our manuscript accordingly. More details find attached in the document. 

Best regards 

Round 3

Reviewer 4 Report

Dear authors

Explain the relevance of the key informants of the qualitative part.

The results of the qualitative part are not coded (uncoded individual texts)

The conclusions are still very weak (nothing has been modified since the previous version) and without logical connection with the entire research process and its results, they still need to be improved.

Best regards